# HYPERSPHERICAL EMBEDDING FOR NOVEL CLASS CLASSIFICATION

## ABSTRACT

Deep neural networks proved to be useful to learn representations and perform classification on many different modalities of data. Traditional approaches work well on the closed set problem. For learning tasks involving novel classes, known as the open set problem, the metric learning approach has been proposed. However, while promising, common metric learning approaches require pairwise learning, which significantly increases training cost while adding additional challenges. In this paper we present a method in which the similarity of samples projected onto a feature space is enforced by a metric learning approach without requiring pairwise evaluation. We compare our approach against known methods in different datasets, achieving results up to $81\%$ more accurate.

## 1 INTRODUCTION

Humans have the ability to identify many different types of objects, Fields (2016). Even when we are not able to name a certain object, we can tell it's differences from a second object, which contributes to the identification of objects we have never seen before and group them into classes based on prior knowledge. *Metric learning* KAYA & BİLGE (2019) is a well adopted approach that identifies novel classes without fine tuning a model on these classes. The approach applies an optimization strategy, which guarantees that the classes a model has seen during optimization form disjoint clusters on the latent space according to a certain metric distance. Some common approaches that use this strategy are: the triplet loss Schroff et al. (2015); constrative loss Hadsell et al. (2006); prototypical networks Snell et al. (2017); constellation loss Medela & Picon (2020); and matching networks Vinyals et al. (2016), here referred to as distance based learners. Another approach in metric learning is called *similarity learning*, where the model receives pairs of inputs and learns that they are similar if they belong to the same class or dissimilar otherwise, as discussed in Sung et al. (2018). During inference on novel classes, distance based learners use the distance between labeled points of the novel class the model was not optimized upon to obtain a representation in the latent space for the novel class and then calculate the distance between new points and each class representation. When considering similarity based learners, a similarity score is calculated between every (class,query) point pair in order to find the most similar pair.

However while enforcing metric properties on the latent space leverages the model knowledge to novel classes, it requires pairwise learning, which limits the scalability of such approaches given the amount of possible pairs.

In this paper we take into account the normalized softmax loss function(NSL), proposed by Wang et al. (2018), and present how it enforces a latent space that obeys the cosine similarity. Based on this, we then present a methodology to apply the *NSL* to the novel classes classification problem. Considering a trained artificial neural network, we add a new neuron to it's last layer and infer the weights that connect the penultimate layer of the network to this neuron. The connection and the new neuron are used to classify a novel class by using few labeled samples of it. Our approach for the open set problem allows us to classify new classes without fine-tuning the model. Instead, we use the same network parameters the model was optimized upon to classify its seen classes and only add a new neuron along with its inferred connection for new classes. We evaluate state-of-the-art approaches to solve the open set problem against our proposed approach, both in the disjoint and joint scenarios, for different datasets. The experimental results show that our approach outperforms

other metric learning strategies and additionally, induces a more scalable training process, as it does not require pairwise learning, leveraging the open set problem technique to deal with large datasets.

The remainder of this paper is structured as follows. First, it presents some theoretical background at section 2. Our methodology and how to classify new classes is described in section 3. Next, we present the results on the joint and disjoint open set problem in section 4. Moreover, we present the use of the NSL approach in a more complex dataset in the field of botany, in section 4.4. We compare our methods to incremental learning in section 4.5. We present related work and lastly, we conclude in section 6.

## 2 PRELIMINARIES

We are given a training dataset $(x_i, y_i)_{i \in \{1, \dots, n\}}$ where, for all $i$, the input $x_i$ belongs to an input space $\mathcal{X} \subset \mathbb{R}^d$, e.g. the space of images, and the output $y_i$ to an output space $\mathcal{Y} = \{1, 2, \dots, K\}$, the set of class labels, where $K$ is the number of classes. Based on this training set, the aim is to find a classifier $h : \mathcal{X} \to \mathcal{Y}$ which produces a single prediction for each input and generalizes well on unseen samples $x \in \mathcal{X}$. When this classifier is a deep neural network, $h$ can typically be expressed as:

$$h(x) = \max_k \hat{\eta}_k(x)$$

where $\hat{\eta}(x) = (\hat{\eta}_1(x), \dots, \hat{\eta}_K(x))$ is the vector of the estimated class probabilities computed as:

$$\hat{\eta}(x) = \psi(\phi(x))$$

with $\phi : \mathcal{X} \to \mathbb{R}^M$ being a succession of layers allowing to compute an $M$-dimensional feature vector representation $\phi(x)$ for any input image $x \in \mathcal{X}$, and $\psi : \mathbb{R}^M \to \mathbb{R}^K$ being the final classification function, typically composed of a fully connected layer followed by a softmax activation function:

$$\psi_k(z) = \frac{e^{w_k z + b_k}}{\sum_{j=1}^K e^{w_j z + b_j}} \tag{1}$$

### 2.1 THE OPEN SET PROBLEM

A classification problem can be formulated as a closed set or open set problem. In the closed set problem context, the optimization process trains a model to learn features that can classify the samples into classes present in the training set. The approach does not require the identification of classes not present in the training set. This is commonly tackled using the Softmax-cross-entropy loss He et al. (2016),Simonyan & Zisserman (2015),Szegedy et al. (2015). In contrast, in the open set problem we are interested in not only identifying the classes present in the training set, but also to be able to use the model to classify new classes by exploiting properties in the latent space yielded during optimization.

### 2.2 CLASSIFYING NEW CLASSES

When tackling the open set problem, we are interested in optimizing models in which the full knowledge the network obtains during optimization can be exploited for classes outside of the training set. The usual *softmax cross-entropy* approach lacks the ability to extract features that obey this property, as the weights between the penultimate layer and the classification layer $w$ are as important as the representation in the latent space of the penultimate layer $z$ as seen in Eq. 1, and the former is undefined for novel classes. Usual approaches for classifying novel classes are explored in metric learning. Metric learning strategies are interesting as novel classes can be identified. However, current strategies based on pairwise learning can be costly to optimize. We discuss in this paper a strategy to remove pairwise learning and still be able to define novel classes for a model.

### 2.3 NORMALIZED SOFTMAX LOSS

Proposed in Wang et al. (2018), the NSL (Normalized Softmax Loss) is a modification of the *softmax* loss that enforces a cosine similarity metric between classes on the latent space. It enforces the

features $z$ that are projected into the latent space to be contained in a $M$ dimensional hypersphere ($M > 3$) where each region of the sphere contains features belonging to a certain class.

If we look again at the classical softmax equation (Eq. 1), the constraints induced by NSL are:

$$\begin{cases} b_k = 0, \forall k \\ \|w_k\| = 1, \forall k \\ \|z\| = \|\phi(x)\| = S, \forall x \end{cases} \tag{2}$$

and finally

$$\hat{\eta}_k(x) = \psi_k(\phi(x)) = \frac{e^{w_k \phi(x)}}{\sum_{j=1}^{K} e^{w_j \phi(x)}} = \frac{e^{S.cos(w_k, \phi(x))}}{\sum_{j=1}^{K} e^{S.cos(w_j, \phi(x))}}$$

where $cos(u, v) = u.v/(\|u\|.\|v\|)$ is the cosine similarity, i.e. the cosinus of the angle between two vectors $u$ and $v$. Note that the hyper-parameter $S$ acts as a temperature of the normalized softmax allowing to control the degree of concentration of the output probabilities $\hat{\eta}_k(x)$.

A geometrical representation that shows the relationship between the weights and the feature vectors obtained with NSL is shown in Figure 2. One can see that the barycenter of the feature vector is aligned with it's corresponding class weights.

## 3 PROPOSED METHODOLOGY

In this paper we aim to compare pairwise strategies, commonly used in metric learning, against the normalized *softmax* loss approach for the open set problem. In this manner we consider both the problem where during inference seen and unseen classes are disjoint, as well as the scenario where the model must identify both the seen and unseen classes together.

More formally, once the network has been trained, we would like to extend the output space to a new set of classes $\mathcal{Y}^* = \{K+1, \ldots, K+K^*\}$ for which we have only one or very few samples $(x_i^*, y_i^*)_{i \in \{1, \ldots, n*\}}$. In particular, we would like to obtain a new classifier $h^* : \mathcal{X} \to \mathcal{Y}^*$ (disjoint scenario) or a new classifier $h' : \mathcal{X} \to \mathcal{Y} \bigcup \mathcal{Y}^*$ (joint scenario). Note that, whatever the scenario, we consider that the function $\phi$ is fixed as well as the pre-trained weights of the seen classes $w_k, \forall k \in \{1, \ldots, K\}$.

### 3.1 CLASSIFYING NEW CLASSES VIA NSL

Given that the function $\phi$ and the weights $w_k$ of the seen classes are fixed, our objective is reduced to optimizing the weights $w_k^*, \forall k \in \{1, \ldots, K^*\}$ of the unseen classes. Using the cross-entropy as the objective function, this can be expressed as:

$$\underset{w_1^*, \ldots, w_{K^*}^*}{\arg\min} \sum_{i=1}^{n^*} -log(\hat{\eta}_{y_i^*}(x_i^*))$$

$$\underset{w_1^*, \ldots, w_{K^*}^*}{\arg\min} \sum_{i=1}^{n^*} -log \frac{e^{w_{y_i^*}^* \phi(x_i^*)}}{\sum_{j=1}^{K} e^{w_j \phi(x_i^*)} + \sum_{j=1}^{K^*} e^{w_j^* \phi(x_i^*)}}$$

In the particular case where we have only one new class (i.e. $K^* = 1$), this simplifies to:

$$\underset{w_1^*}{\arg\max} \sum_{i=1}^{n^*} w_1^* \phi(x_i^*) = \underset{w_1^*}{\arg\max} \ w_1^* \sum_{i=1}^{n^*} \phi(x_i^*)$$

which leads, with the constraints of Eq. 2, to:

$$w_1^* = \frac{1}{n^*} \sum_{i=1}^{n^*} \frac{\phi(x_i^*)}{\|\phi(x_i^*)\|} = \frac{1}{S.n^*} \sum_{i=1}^{n^*} \phi(x_i^*) \tag{3}$$

The weight $w_1^*$ of a new class can thus simply be computed by averaging the feature vectors of the images $x_i^*$ of the new class. This simple theoretical result does not hold anymore when there is more than one novel classes (i.e. when $K^* > 1$). However, as we we will see in our experiments, using

this estimation procedure for other new classes provides a good approximation of the exact optimal weights and is quite effective in practice. More formally, we propose to estimate the weights $w_k^*$ of each of $K^*$ new classes as:

$$w_k^* = \frac{1}{S} \frac{\sum_{i=1}^{n^*} \phi(x_i^*) \mathbb{1}(y_i^* = K + k)}{\sum_{i=1}^{n^*} \mathbb{1}(y_i^* = K + k)} \tag{4}$$

In the *joint scenario*, we are interested in a classifier on both the seen classes and the new classes. This can be expressed as:

$$h_{joint}(x) = \sum_{i=1}^{n^*} \max_k \frac{e^{w_k^* \phi(x_i^*)}}{\sum_{j=1}^{K} e^{w_j \phi(x_i^*)} + \sum_{j=1}^{K^*} e^{w_j^* \phi(x_i^*)}} \tag{5}$$

where the $w_j$ and $\phi()$ are pre-trained on the seen classes and the new weights $w_k^*$ are computed with Eq. 4.

In the *disjoint scenario*, we are interested in a classifier on the new classes only (in a transfer learning way):

$$h_{disjoint}(x) = \sum_{i=1}^{n^*} \max_k \frac{e^{w_k^* \phi(x_i^*)}}{\sum_{j=1}^{K^*} e^{w_j^* \phi(x_i^*)}} \tag{6}$$

where $\phi()$ is pre-trained on the seen classes and the new weights $w_j^*$ are computed with Equ. 4.

A dataflow depicting our approach to infer the weights for novel classes is presented in Figure 1.

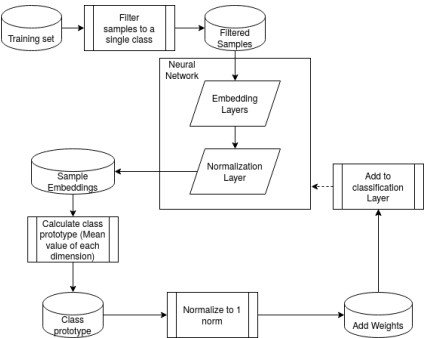

Figure 1: Diagram presenting the approach to infer weights for the decision layer for new classes

## 4 RESULTS

### 4.1 EXPERIMENTAL SETUP

In this section we present the experimental setup. All experiments presented for the FASHION MNIST and CIFAR datasets were performed using google collaboratory. Experiments using the Pl@ntNet dataset were performed using a Dell PowerEdge R730 server, with 2 CPUs Intel (R) Xeon (R) CPU E5-2690 v3 @ 2.60GHz; 768 GB of RAM; and running on a Linux CentOS 7.7.1908 kernel version 3.10.0-1062.4.3.e17.x86_64. The machine is equipped with a single NVIDIA Pascal P100 GPU, with 16GB RAM. Implementations were performed using Python 3.7 along with the Keras deep learning library.

### 4.2 EVALUATING THE DISJOINT SCENARIO

In this section we show the results of evaluating the model accuracy on the test set from seen and unseen classes by employing a VGG based model with two convolutional blocks.

We optimize the model on $K = 10 - K^*$ seen classes, and we use the trained network to classify the $K^*$ unseen classes, without considering the seen ones as possible answers. The approach to do

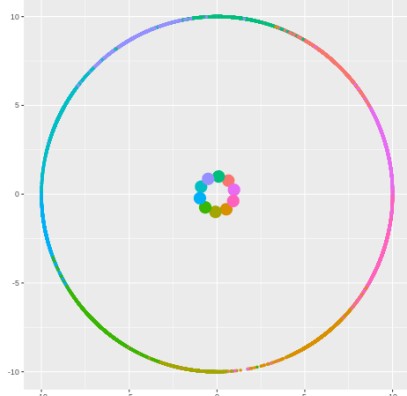

Figure 2: Embedding obtained on the cifar10 dataset when using a latent space with two dimensions using NSL, each color represents a different class. Inner points are the class weights while outer points come from the training set, notice how classes from the outer circle are aligned with the inner circle

| $K^*$ | NSL | Triplet | Constrative |
|---|---|---|---|
| 2 | **0.852** | 0.663 | 0.700 |
| 3 | **0.725** | 0.570 | 0.551 |
| 4 | **0.629** | 0.406 | 0.422 |
| 5 | **0.545** | 0.296 | 0.328 |

Table 1: Model results for the cifar 10 dataset. $K^*$ refers to the amount of unseen classes, while other columns refer to the method and accuracy obtained on the test set.

| N | NSL | Triplet | Constrative |
|---|---|---|---|
| 2 | **0.897** | 0.62 | 0.703 |
| 3 | **0.876** | 0.39 | 0.469 |
| 4 | **0.841** | 0.27 | 0.312 |
| 5 | **0.807** | 0.22 | 0.2 |

Table 2: Model results for the Fashion Mnist dataset. $K^*$ refers to the amount of unseen classes while other columns refer to the method and accuracy obtained on the test set.

so was presented in Eq. 6, in section Proposed Methodology. Results are presented in Tables 1 and 2, where we compare the adoption of the NSL approach against two metric learning strategies in a disjoint settings. In the first line of the table, we present a scenario in which we train with 8 random classes and evaluate on the other two unseen classes. In the second line, the experiment trained the model with 7 classes and tested with 3 unseen ones, and so forth. Our results show that in both datasets the NSL outperformed those metric learning strategies for evaluating novel classes in a disjoint scenario. To evaluate both the triplet loss as well as the constrative loss methods, we first built the embedding representation, and then fed this representation into a k-nearest neighbours model trained on the average embedding of the class, using the same number of samples as NSL.

### 4.3 EVALUATING THE JOINT SCENARIO

In this section, we present the results when the novel classes must be integrated into the classification process along with the classes used for optimization. To this end, the function that we want to optimize is described in Eq. 5. The model is optimized with $10 - K^*$ classes and we evaluate the accuracy on these and on the $K^*$ unseen classes considering 10 total classes. Results are presented in Tables 3 and 4.

Tables 3 and 4 depict the results of comparing our approach using $NSL$ with metric learning strategies: Triplet loss and Constrative loss, on both Cifar and Fashion Mnist datasets, considering a joint

|  | Seen | | | Unseen | | |
|---|---|---|---|---|---|---|
| $K^*$ | NSL | Triplet | Constrative | NSL | Triplet | Constrative |
| 2 | **0.559** | 0.195 | 0.205 | **0.501** | 0.226 | 0.189 |
| 3 | **0.578** | 0.128 | 0.182 | **0.433** | 0.156 | 0.176 |
| 4 | **0.620** | 0.082 | 0.166 | **0.391** | 0.127 | 0.160 |
| 5 | **0.641** | 0.05 | 0.215 | **0.357** | 0.146 | 0.176 |

Table 3: Model results for the Cifar10 dataset. $K^*$ refers to the amount of unseen classes while other columns refer to the method and accuracy obtained on the test set. Seen refers to the accuracy in the $10 - K^*$ classes while unseen on the $K^*$ classes

|  | Seen | | | Unseen | | |
|---|---|---|---|---|---|---|
| $K^*$ | NSL | Triplet | Constrative | NSL | Triplet | Constrative |
| 2 | **0.854** | 0.708 | 0.657 | **0.773** | 0.724 | 0.613 |
| 3 | **0.842** | 0.705 | 0.440 | **0.771** | 0.716 | 0.390 |
| 4 | **0.860** | 0.688 | 0.284 | **0.739** | 0.680 | 0.286 |
| 5 | **0.878** | 0.690 | 0.190 | **0.719** | 0.691 | 0.181 |

Table 4: Model results for the fashion mnist dataset. $K^*$ refers to the amount of unseen classes while other columns refer to the method and accuracy obtained on the test set. Seen refers to the accuracy in the $10 - K^*$ classes while Unseen on the $K^*$ classes

scenario. The NSL approach outperformed both metric learning strategies on the two evaluated datasets, including seen and unseen classes predictions.

### 4.4 CASE STUDY: THE PL@NTNET DATASET

In order to assess the NSL approach on real world data, we evaluate it under the closed and open set problems on a dataset built from *Pl@ntnet* database. Pl@ntnet is one of the largest citizen science observatory in the world relying on a mobile application Affouard et al. (2017) that allows contributors to identify plants using their smartphone (an application that uses convolutional neural networks). The task is challenging as available pictures have different levels of quality, as well as multiple species from many different parts of the world as shown in Figure 3.

The problem becomes relevant to the evaluation of the proposed approach given that the scenario is usually represented by a long tail distribution, in which some classes are very common, while others are rare and lack significant available training data.

The subset of the Pl@ntnet data used was obtained from Garcin (2021) and has a total of 182 classes.

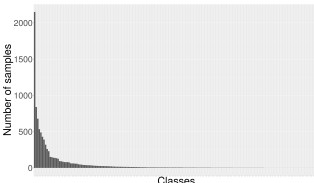

Figure 3: Distribution of the training dataset. Note the long tail distribution presenting how there are many classes with small amounts of data and few with a large amount.

#### 4.4.1 EXPERIMENTAL DESIGN

As it is clear from Figure 3, there is a high imbalance among classes in the Plantnet dataset. Thus, there are many classes in the training set which have very small amounts of data. Since many plant species have few samples, we are interested in exploring the performance of NSL, where a model is optimized only on more common species and weights to classify uncommon species are inferred, as discussed in section 3.1. To this end, we perform experiments by optimizing the model only on classes where the number of samples is larger or equal to $N = \{200, 100, 50, 25\}$, which results

| Number of classes | 10 | 16 | 28 | 43 |
|---|---|---|---|---|
| NSL accuracy | 0.7349 | 0.6617 | 0.5382 | 0.3974 |

Table 5: Model accuracy on the test set optimized for 100 epochs on weighted cross-entropy on seen classes.

in $K = \{10, 16, 28, 43\}$, where $K$ is the number of classes, and present the results for joint and disjoint settings. Unseen classes will be selected randomly among those with number of samples $M$, $M < N$, and results will be presented with the average of 30 runs. All models are optimized on 100 epochs and weights that minimize validation loss are used for inference.

To describe the model architecture, first we define a convolutional block as two convolutional layers with same kernel size followed by a maxpooling layer. The model is structured with four convolutional blocks as previously defined in sequence, with 3x3 kernels of sizes (64,64),(256,256),(256,512),(512,1024). Following the four convolutional blocks we add a flattening layer and feed its output to a fully connected layer with 1024 neurons and no activation function. The output is then normalized to $S$ norm and fed to the classifier. The Pre-processing step prepares the data by normalizing it to $[0, 1]$ range and reshaping it to a $< 96, 96, 3 >$ shape. Models are optimized with weighted cross-entropy by passing the class weights arguments to *keras fit* function to deal with unbalanced classes. For the open set tasks, we report on balanced accuracy to better take into account class imbalance.

### 4.4.2 RESULTS

In this section, we present results using balanced accuracy for a set of models built with varying number of classes in the last layer. As already discussed, we selected seen classes based on a filter on the number of samples $M \geq N$. Under this constraint, the relation between the number of samples and the number of classes is as follows: $[25, 43], [50, 28], [100, 16], [200, 10]$.

In Table 5, we present the model accuracy for the four different models to be used on the Plantnet experiment. All models were optimized by receiving the same amount of samples per epoch, as well as the number of epochs. This set of models is then used to evaluate both the disjoint and joint scenarios.

### 4.4.3 DISJOINT SCENARIO

In this subsection, we present the disjoint analysis for the Plantnet dataset. As discussed in section 3.1, in our approach, we instantiate the model trained with seen data, without the last layer. We pass the samples from the unseen training set through the network and obtain their enbeddings from the penultimate network layer. Next, we compute the weights, using the enbeddings, according to Eq. 3. Observe that we adopt a random selection of unseen classes to compute weights and to perform inference. We present the results of 30 runs, for each value of $K^*$, for novel classes inference using balanced accuracy. Results are presented in Figure 4.

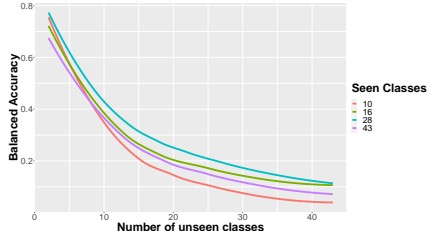

Figure 4: Comparing models sharing the same architecture and optimized upon the same data and number of epochs, but with different amount of seen classes. We evaluate their ability to classify novel classes.

In Figure 4, we show a comparison among models trained on a different number of classes with the same amount of data, by presenting their ability to identify novel classes in a disjoint scenario. Our results show that the diversity of classes seen during training allowed the model to show robustness

for novel classes. This is observed by the worst results in Figure 4, obtained for the inference of novel classes, when training with only 10 seen classes. However it is also important to note that optimizing on a higher number of classes is a more complex problem, requiring more data to be seen by the model, more updates or a more complex model to learn robust features for all seen classes. This is shown on the curve obtained from the model optimized upon 43 classes, which ranks the second worst. Our best result was seen on the model with 28 classes, which had a high class diversity while also learning robust features during training.

### 4.4.4 JOINT SCENARIO

In this subsection, we present the results of the joint scenario for the Pl@ntnet dataset. We instantiate the model trained on $K$ classes. Then, we add other $K^*$ unseen classes, so that the model must classify between $K + K^*$ classes. Weights for the $K^*$ classes are inferred as described in section 3.1. We report the model accuracy for all classes, seen and unseen, as well as the model behavior for only unseen classes. Results are presented in Figures 5 and 6.

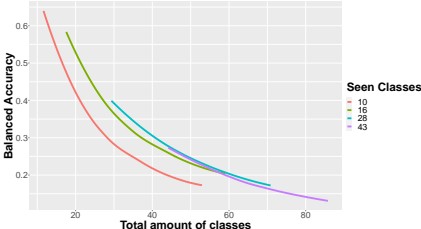

Figure 5: Analysis for the joint scenario showing the results for different models on seen and unseen classes.

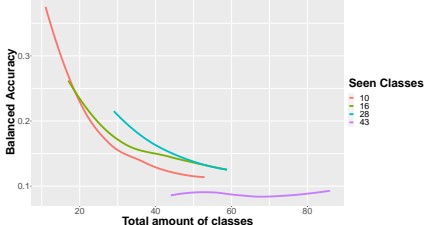

Figure 6: Analysis for the joint scenario showing the results for different models on the unseen classes.

In Figures 5 and 6 we present our results for the joint scenario evaluating the overall model quality as new classes are added as well as showing the balanced accuracy calculated only on the unseen classes. Our conclusions on the disjoint scenario, seen in Figure 4, also hold in these results. We can see that for both of these cases the model trained on 28 classes has the higher balanced accuracy, given the same number of total classes. Moreover, the model optimized on 43 classes shows the worse balanced accuracy. The lack of diversity of the model optimized on 10 classes influences it's quality as the number of novel classes increases in both scenarios.

### 4.5 FEW SHOT SCENARIO FOR INCREMENTAL LEARNING

In Eq. 3, we show how our proposed methodology for inferring weights actually finds the set of weights that minimizes cross-entropy, whenever a single novel class is included. However, when including multiple classes, our proposal may not yield the optimum set of weights for each new neuron. In this section we present a set of experiments comparing the performances obtained by our inferred weights with the ones obtained through incremental learning, i.e. by minimizing the cross-entropy loss on samples of the new classes while freezing all the other network weights. Experiments were performed using the Cifar10 dataset. The initial model is trained on the training samples of $K$ seen classes and the incremental learning phase is computed on the training samples of $K^*$ unseen classes (while freezing all the other network weights).

To evaluate the proposed strategy in a few shot scenario, we train the Resnet50 architecture using the NSL constraint, as provided by keras, on a subset of classes of the cifar10 dataset. Once the network has been trained for 100 epochs on a subset of the dataset, we sample a small number of examples of the classes that were unseen during training in a few shot scenario (one, five and twenty five shots). We infer the weights for the novel classes using the methodology described in section 3.1 and measure the model quality on the test set compared to the incremental learning approach (using the same few shots for each class). We report the results for one, five and twenty five shot scenarios on the CIFAR10 dataset. Figure 7 depicts similar results for all scenarios, considering the accuracy of predictions. Note that when the number of novel classes is smaller or equal to the number of seen classes our approach tends to achieve higher accuracy.

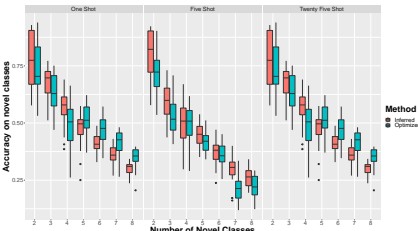

Figure 7: Accuracy obtained in different few shot scenarios for the cifar10 dataset when comparing using inferred weights versus further optimizing them.

## 5 RELATED WORK

Creating models that are able to classify novel classes is a task that is explored in different fields of artificial intelligence. There are two fields our work falls upon. The first is called metric learning discussed in section 2.2, while the second is called incremental learning briefly presented in section 4.5.

In metric learning, which is a sub-field of few shot learning, one aims to train a model to identify classes via some property in a metric space, enforced during training. The enforced property can be for example that examples of the same class form a cluster according to some predefined metric and each class has its own cluster. These properties can then be explored to identify novel classes given that it is possible to determine the cluster of a novel class with some labeled examples without retraining the model. There are many works that fall in this category Schroff et al. (2015),Hadsell et al. (2006),Medela & Picon (2020),Vinyals et al. (2016). While all these approaches enforce metric properties on the latent space, they also require pairwise training, which our approach does not require.

## 6 CONCLUSION

In this paper, we presented how the normalized softmax loss can be employed on the open set problem. We presented results on different datasets for both the disjoint and joint open set problems and compared them to metric learning strategies. We show that the NSL based approach demonstrates superior results producing more robust features and implementing a less costly optimization procedure, as it does not require pairwise training. Results on a real world use case evaluating on a subset of the Pl@ntnet data shows how our approach can be employed to identify classes unseen during optimization, with weights associated to the classification of new data inferred by the approach.

## 7 ACKNOWLEDGMENTS

The authors would like to thank Petrobras for supporting this work through the project "Development of an Intelligent software platform". We would also like to thank the INRIA-Brazil Associated Team cooperation project HPDaSc.

## 8 REPRODUCIBILITY STATEMENT

In this section we detail the steps we ensured to ensure that our work is reproducible. To ensure data availability we mostly use public datasets that are available in the keras.datasets interface. The subset of the Pl@ntnet dataset we used in this paper available as numpy arrays in the plantnet folder that is available via a google drive link that is presented in the appendix section.

Regarding data pre-processing, all pre-processing steps as well as the structure of the models are presented in the main paper.

Concerning the mathematical formulation of the problem, the main mathematical formulation is presented in the main paper while additional information about the area of the problem is presented in the appendix on metric learning and the latent space.

Lastly regarding experiment reproducibility, all experiments were organized into jupyter notebooks and these are organized into an folder that is available via a google drive link on the appendix.

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

## A    APPENDIX

The following sections contain additional analysis and theoretical discussion that is not integral to the understanding of the paper but the authors would like to share about the research performed.

### A.1    METRIC LEARNING AND THE LATENT SPACE

Deep neural networks learn a set of transformations and relations among the inputs in order to obtain the desired output during optimization. The network can be broken into two parts: the projection module ($\phi(x)$), which takes the data and transforms it into a representation in the latent space; and the data representation processing, applied by the layer implementing the desired task ($\psi(z)$). The latter enforces the latent space to have some properties that are defined by the task. When the network is trained with a cross-entropy loss, the objective is the following:

$$\arg\min_{\theta} \sum_{i=1}^{n} -log(\hat{\eta}_{y_i}(x_i))$$

where $\theta$ is the set of all parameters of the network (for both $\psi(x)$ and $\phi(x)$). Thus, after optimization, we generally have that $\hat{\eta}_{y_i}(x_i) >> \hat{\eta}_j(x_i)$ for $j \neq y_i$ which can only be achieved if: $w_{y_i}\phi(x_i) + b_{y_i} >> w_j\phi(x_i) + b_j$ for $j \neq y_i$. In other words, the *softmax cross-entropy* approach enforces the inequality $w_i z_i + b_i >> w_j z_i + b_j$ in the latent space, where $i, j$ represent different classes Wang et al. (2018). A proposed alternative that optimizes the latent space directly and can enforce metric properties that allows the model to be used for novel classes is know as *metric learning*.

The metric learning approach learns a set of features that obey a metric distance on the latent space. The model can be optimized to learn a similarity metric between pairs, as proposed in Sung et al. (2018), or can enforce the latent space to obey a predefined metric distance like euclidean distance or cosine similarity. Some strategies, such as the *constrative loss* Hadsell et al. (2006), learn on pairs of data, while others learn using triplets like *triplet loss* Schroff et al. (2015).

Optimization on these approaches aim to obtain disjoint clusters for each class of interest in the latent space, according to a predefined metric distance. As a desired consequence of the approach, classification can be performed for novel classes by using the representation of an anchor example and calculating the metric distance between a query point and the anchor.

### A.2    HOW DOES THE AMOUNT OF SAMPLES AFFECT THE CLASS PROTOTYPE ?

A common scenario in which the identification of unseen classes appears is the one where the amount of available data samples for classes of interest is small or the cost of optimizing another

model to include the new classes becomes too costly. Therefore, all strategies discussed in this paper classify new classes based on labeled examples without retraining. The influence of the number of samples needed to perform classification tasks using the NSL approach is shown in three different datasets in Figure 8 for a model optimized for 30 epochs. We use keras base learning rate with adam optimizer. The experiment consider that the model was trained on all ten classes. Models for mnist and fashion mnist consider only dense layers, while cifar considers two convolutional blocks with 32 and 64 filters each. We create the class prototype using the inferred weights obtained via Eq 4, an strategy similar to Snell et al. (2017).

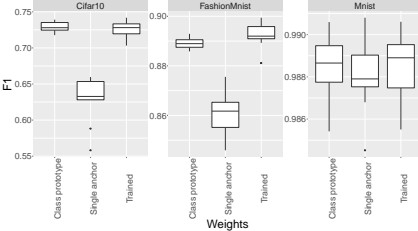

Figure 8: F1 score on the test set for three different datasets where we show results when weights are: (a) Trained: the ones found during optimization; (b) Single anchor: weights are inferred using a single random anchor example; and (c) weights are inferred using the training set to build a class prototype.

As it can be seen on Figure 8, weights inferred using Eq. 3 on the training set maintain the same model accuracy as by using the weights obtained during optimization. We also can observe that model quality, when inferring via a single example, decays in relation to the task complexity. On the x axis we present how the class weights were obtained where class prototype uses the whole training set to infer the weights according to our methodology, while single anchor uses a single random example from the training set for weight inference. Y axis presents the F1-Score on the test set.

### A.3 COMPARISON TO INCREMENTAL LEARNING USING MANY SAMPLES

In Eq. 3, we show how our proposed methodology of inferring weights actually finds the set of weights that minimizes cross-entropy, whenever a single novel class is included. However, when including multiple classes, our proposal may not yield the optimum set of weights for each new neuron. In this section we present a set of experiments comparing the performances obtained by our inferred weights with the ones obtained through incremental learning, i.e. by minimizing the cross-entropy loss on samples of the new classes while freezing all the other network weights. Experiments were performed using the fashion mnist dataset. The initial model is trained on the training samples of $K$ seen classes and the incremental learning phase is computed on the training samples of $K^*$ unseen classes (while freezing all the other network weights).

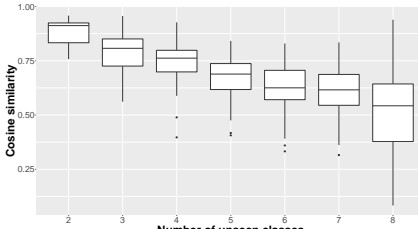

Figure 9: Cosine similarity between inferred weights, obtained as in Eq. 3, and optimized weights, obtained through cross-entropy optimization using the incremental learning constraint, for different numbers of novel classes on the fashion MNIST dataset

Figure 11 first shows the cosine similarity between the inferred and the optimized weights for different numbers of unseen class. As we can observe, when the number of novel classes is small, the two

sets of weights are almost identical, which means that the inferred weights are as good as the ones optimized through incremental learning (while being much faster and simpler to compute). With larger numbers of novel classes, we can observe that the mean cosine similarity is still very high. This suggests that the gain of incremental learning might not be very high in this case as well.

To quantify this gain, Figure 10 presents the ratio of the accuracy achieved through incremental learning over the one obtained with the inferred weights (using MNIST test set). We see that when including small number of novel classes the ratio stays close to 1, showing no strong accuracy improvement due to optimization. When including more classes, the gain of incremental learning can be higher (up to 1.55 for 8 unseen classes) but this requires 4-5 epochs on the training set. This suggests that the inferred weights may be used to initialize the incremental phase and get a faster convergence when having a lot of data for the novel class. In the next section we perform the same experiment considering small data.

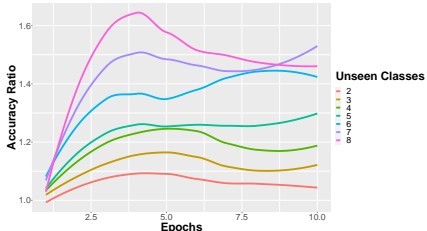

Figure 10: Ratio of accuracy after optimization versus inferred weights on $unseen = 10 - seen$ classes when optimization occurs on seen classes for the fashion mnist dataset.

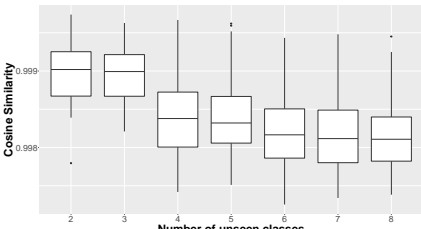

Figure 11: Cosine similarity between inferred weights, obtained as in Eq. 3, and optimized weights, obtained through cross-entropy optimization using the incremental learning constraint, for different numbers of novel classes on the fashion MNIST dataset when using only two samples to generate the novel class

## B    EXPERIMENTS

All performed experiments are available via the following google drive folder: `https://drive.google.com/drive/folders/1P2WUw11k9s1IbSdqT38nNbM61m6KQyfD?usp=sharing`

The subset of the plantnet data is already available in numpy array format inside the plantnet folder in this link

### B.1    NSL ARCHITECTURAL SEARCH ON THE CLOSED SET

In this section, we present in Table 6 an ablation study comparing NSL against softmax loss applied to the same network architecture. We define the following deep neural networks: an Inception block, as a single Inception module as proposed in Szegedy et al. (2015); a VGG block is a composition of two Convolutional layers with 3x3 kernel and a 2x2 MaxPooling layer; while a Resnet block is built with a 3x3 convolution with a residual connection. For the number of filters we increase them in the order $< 32, 64, 128 >$ values.

| Architecture | Number of blocks | SL | NSL |
|---|---|---|---|
| Inception | 1 | 0.473 | **0.633** |
| Inception | 2 | 0.47 | **0.746** |
| Inception | 3 | 0.42 | **0.803** |
| ResNet | 1 | 0.1 | **0.610** |
| ResNet | 2 | 0.1 | **0.75** |
| ResNet | 3 | 0.1 | **0.81** |
| VGG | 1 | 0.175 | **0.638** |
| VGG | 2 | 0.187 | **0.738** |
| VGG | 3 | 0.643 | **0.823** |

Table 6: Analysis on the closed set for different architectural setups optimized on softmax vs normalized softmax for the cifar10 dataset. SL and NSL present model accuracy on the test set.

In table 6 we compared the normalized softmax loss vs softmax loss on different architectures. We note that for all architecture variations as well as variations in the network depth, the normalized softmax loss outperformed the softmax loss. We also denote that on softmax loss the choice of base architecture (Inception,Resnet,VGG) significantly influenced the model accuracy as much as depth. In contrast, the normalized softmax loss was more influenced by model depth in this dataset.

