# OpenReview forum: "Hyperspherical embedding for novel class classification"
_ICLR.cc/2022/Conference — ICLR 2022 Submitted_

### Official Review · Reviewer_AHvJ · 2021-10-29

**Correctness:** 3
**Technical Novelty And Significance:** 2
**Empirical Novelty And Significance:** 1
**Recommendation:** 3
**Confidence:** 4

**Main Review:**

#### Strengths:
+ The presentation and writing is clear and easy to follow.
+ The proposed approach is simple and efficient.

#### Weaknesses:
- The paper states that (deep) metric learning approaches (DML) suffer from high computitational complexity due to reliance on pairwise learning constraints. However, many approaches in DML effectively circumvent the computational burden using sampling strategies [e.g. A] or class proxies [e.g. B], etc. In particular the latter has shown to be both very computationally efficient and competitive with the state-of-the-art in DML.

- In general lots of references to established works in Deep Metric Learning (DML) and Few-Shot Learning (FSL) are missing. For extensive discussions of both fields, e.g. see [C, D].

- The core idea of the presented approach is adding additional neurons, respectively class weights, for new classes to the entropy-loss based classification. The class weights are set to the average over inferred weights based on few representatives of the new classes and subsequently fixed for classifiying unseen samples. However, it is questionable if the proposed approach would work on larger, more complex and finegrained datasets, as inferring class weights by averaging over few samples may be too unstable for representing fine-grained class features. More experiments on larger, standard benchmark sets would be very helpful.

- The experimental section lacks details about the trained model, i.e. architecture details and optimization paramters. No standard architectures commonly used in DML or FSL ([C,D]) seem to be considered for fair comparison to the standard/State-of-the-art approachess. Moreover, comparison to state-of-the-art approaches in DML,FSL are lacking in the quantative evaluation. Only Tripet and Contrastive Learning seems to be considered – however, reference about the exact implementation details of these approaches are missing as well. Finally, the evaluation also does neither consider standard benchmark sets in DML (e.g. CUB200, CARS196, Standford Online Products) or FSL (Mini-ImageNet, CIFAR100), nor follows the established evaluation protocols of these areas. In total, the evaluation protocol significantly lacks comparison to established results in DML, FSL and, thus, does not allow for proper evaluation of the effectiveness of the proposed approach.

- Evaluation on the PlantNet dataset lacks comparison to other approaches/baselines.

---------------
References:

[A] Wu et al. 2017; Sampling Matters in Deep Embedding Learning

[B] Kim et al 2020; Proxy Anchor Loss for Deep Metric Learning

[C] Roth et al. 2020; Revisiting Training Strategies and Generalization Performance in Deep Metric Learning

[D] Tian et al. 2020; Rethinking Few-Shot Image Classification: a Good Embedding Is All You Need?

**Summary Of The Paper:**

The paper proposes an approach to Few-shot Learning based on the the CosFace (Normalized Softmax) Loss. After pretraining, class weights are added to the cross entropy loss for each new class in the test set, which are computed by averaging over inferred weights from a support set while fixing the remaining network weights (feature extractor). Experiments conducted on CIFAR10 and Fashion-MNIST indicated gains over baseline approaches.

**Summary Of The Review:**

The paper lacks novelty and significance both for the presented approach and the empirical results. Further, the quantitative evaluation is based on non-standard datasets, evaluation protocols, architectures and baselines, hence does not provide a proper basis to evaluate the effectiveness of the proposed approach.

---

### Official Review · Reviewer_Tp5p · 2021-11-02

**Correctness:** 3
**Technical Novelty And Significance:** 1
**Empirical Novelty And Significance:** 2
**Recommendation:** 3
**Confidence:** 4

**Main Review:**

This paper presents a good empirical study on dataset showing the effect of the number of class seen and unseen. However, there is a concern regarding some statement made by the author and regarding the novelty presented in this work.

Strength:
- Case study in sec 4.4, this case study gives valuable insight on the effect of the data distribution on learning. Fig 4,5 and 6 shows that the choice of seen classes is important for learning on unseen classes and that inter-class diversity is more important than having a high number of classes.

Weakness:
 - In the abstract, the author states that the results are 81 % more accurate but do not specify compare to what. I do not see in the table a jump in performance of more than 81% compare to the proposed baselines.
- The authors chose to have metric learning methods as a baseline and compare their classifier with triplet/contrastive networks. While deep metric learning advanced, recent methods proposed frameworks that do not necessary need to form pairs.
See "Proxy anchor loss for deep metric learning from S Kim et al" or "No fuss distance metric learning using proxies by Y Movshovitz-Attias et al". The method proposed by the authors seems to be an easier setting than the method proposed in the stated paper since they are classifying in a zero-shot manner (the new classes are not seen in training).
- The main issue that could bring concern is the design of the weight for the new class in Eq 4. By not fine-tuning the network on the new accessible classes, there is a chance that the features used to compute the new weights belong to a similar distribution than features from another existing classes. It could then happen that a new weights is equal to an already existing weight. How to make sure that it does not happen ?
- Another issue is the lack of comparison with existing work. It would have been interesting to compare with state-of-the art incremental learning method or the newest metric learning method. It would have provided a strong support for claiming that the authors' work is new and performant.
- There are too few related work reviewed in Sec 5.

**Summary Of The Paper:**

This paper is about classification of images in an open set setting. Data coming from new classes are introduced to the network after training on data from a fixed set of known classes. The goal is to be able to correctly classify the old and new classes either jointly or not. This paper proposes a method for handling new classes by increasing the size of the classifier weights for each new classes. The network is trained using the Normalized softmax loss and new classifier elements are added by finding the center of mass of the data coming from the new set of class. The method performs on FASHION MNIST, CIFAR and Plantnet datasets.

**Summary Of The Review:**

My recommendation for this paper is a strong reject. I am not convinced of the novelty proposed in this paper and there are too few comparison with existing work to strengthen the validity of the performance on the proposed datasets.

---

### Official Review · Reviewer_t3Uk · 2021-11-02

**Correctness:** 2
**Technical Novelty And Significance:** 2
**Empirical Novelty And Significance:** 2
**Recommendation:** 3
**Confidence:** 4

**Details Of Ethics Concerns:**

No concerns.

**Main Review:**

1) The paper completely missed the related works. Many papers addressing this problem are ignored. Basically only two metric learning approaches are introduced. The paper should improve the related work following this recent survey:
Geng, Chuanxing, Sheng-jun Huang, and Songcan Chen. "Recent advances in open set recognition: A survey." IEEE transactions on pattern analysis and machine intelligence (2020).

2) In the reviewer's opinion the paper in its current shape is not ready to be published. The proposed method is loosely motivated and the comparisons are somewhat limited.

3) Paper contributions are not highlighted, difficult for a reviewer to understand the novelty of the paper if any.


**Summary Of The Paper:**

The paper presents how the normalized softmax loss can be used on the open set problem. Superior results seem to be achieved without the pairwise pairwise training

**Summary Of The Review:**

Paper difficult to review in its current shape.

---

### Official Review · Reviewer_qMdp · 2021-11-02

**Correctness:** 3
**Technical Novelty And Significance:** 2
**Empirical Novelty And Significance:** 2
**Recommendation:** 3
**Confidence:** 3

**Main Review:**


This work follows on from the Normalized sotmax loss function work by Wang et al 2018.
In that work, the last layer of a network learns feature vectors that are constrained/projected to a unit sphere.
Learned simultaneously are weight vectors (w) for each class.

This work considers the arrival of a new set of examples for a set of new (previously unseen) classes.
These examples come with labels, but there are assumed to be few examples and  we would like to lever the features from the "pretraining" of the network on the original training set.

Ideally, one would use the cross entropy loss to optimize a new set of classifying vectors (w) in the unit sphere for the new classes. The authors point out that for one class, this is equivalent to making the new class vector (w) to be the normalized average of all the feature vectors for the new class.
(Note: the weight vectors for the classes and the features vectors share the same space in the unit sphere.)
Then the authors consider what might happen if they apply the same method for more than one new class. This has less theoretical justification. In effect for each new class, a weight vector is generated by computing the normalized average of the feature vectors for examples from that class.

They consider different settings - joint and disjoint, and evaluate on a few different datasets.
The main results (tables 1,2,3,4) appear impressive, but I have some reservations..

1) It is not clear what the feature dimensions are for the datasets.
2) There is little detail about the triplet/contrastive methods used in the comparison.
3) I would expect to compare against many more competing methods for this proposal.

The first three evaluation datasets were MINST, Cifar10 and Fashion MNIST.
A lot of space in the paper is given to The Pl@ntnet dataset. (4.4.*)
It is a careful, appropriate and novel application of the method.
There is a discussion about how accuracy is affected for varied numbers of unseen classes under disjoint and joint scenarios with different training / unseen class splits. It is revealed that it is helpful to pre-train with a large number of seen classes, but that too many "seen" classes can be detrimental. So there is a trade of for this parameter, for this dataset at least.
This work (pl@ntnet) does not have any baselines or comparisons with other methods, it shows that the method can be applied but doesn't tell us much else - there are no strong conclusions or detailed analysis of the results.

In section 4.5 the authors compare their method to incremental learning. (In this case by training a new layer to the new classes while
freezing all other weights). This would be a section of great interest to anyone reading this paper but it is short, and the results are
 presented in one figure which is hard to read and no strong conclusions can be drawn from it.


**Summary Of The Paper:**

This paper proposes a method to repurpose classifiers trained using the Normalized Softmax Loss function to novel classes by averaging the examples from the same previously unseen class to gain a new weight vector for that class.
The method is simple and effective, even though it is not theoretically justified for more than one class.


**Summary Of The Review:**

I like the method - it is simple and intuitive, and the initial results appear impressive.
But for such a simple model the burden of proof is high, and there should be many more detailed comparisons with SOA zero-shot methods allowing the reader to compare and contrast benefits.
So I would like to see more detailed analysis on datasets and methods, for example, I would also like to see study on:
1) how does the dimension of the feature vector affects performance?
2) what happens when a set of new classes are very similar to a known class?
Perhaps some synthetic data would allow to demonstrate the strengths and weaknesses of the method.

---

### Decision · Program_Chairs · 2022-01-20

**Decision:**

Reject

**Comment:**

This paper tackles an open-set setting where new classes (with few labeled examples) are introduced after the initial pre-training on different categories. A simple approach is proposed based on a normalized softmax classifier and feature averaging to generate a classifier for the new categories. Results are shown on a few standard datasets as well as the Pl@ntnet dataset.

While reviewers found the topic and setting (as well as Pl@ntnet dataset) interesting, they had significant concerns on the novelty (t3Uk, Tp5p, AHvJ), contribution, and rigor of the empirical evaluation. Since the method is simple and largely leverages from prior works, the latter is especially important; reviewers pointed out that some of the latest in metric learning is ignored (e.g. Proxy Anchor, Tp5p and AHvJ), and no comparison is made to other classes of methods that (by the authors' admission) are very close to the setting such as open-set recognition (especially those that seek to classify new categories) and incremental learning.

Unfortunately, no rebuttal was provided by the authors, so these significant concerns remain and the paper cannot be accepted as-is. Since the reviewers did appreciate the setting and dataset, I recommend refining the paper and significantly beefing up the empirical evaluation for future resubmissions.